# Evaluating a Nationally Localized AI Chatbot for Personalized Primary Care Guidance: Insights from the HomeDOCtor Deployment in Slovenia

**DOI:** 10.3390/healthcare13151843

**Published:** 2025-07-29

**Authors:** Matjaž Gams, Tadej Horvat, Žiga Kolar, Primož Kocuvan, Kostadin Mishev, Monika Simjanoska Misheva

**Affiliations:** 1Department of Intelligent Systems, Jožef Stefan Institute, 1000 Ljubljana, Slovenia; tadej.horvat@ijs.si (T.H.); ziga.kolar@ijs.si (Ž.K.); primoz.kocuvan@ijs.si (P.K.); 2Applied Artificial Intelligence, Alma Mater Europaea University, Slovenska Street 17, 2000 Maribor, Slovenia; 3Faculty of Computer Science and Engineering, Ss. Cyril and Methodius University in Skopje, 1000 Skopje, North Macedonia; kostadin.mishev@finki.ukim.mk (K.M.); monika.simjanoska@finki.ukim.mk (M.S.M.)

**Keywords:** conversational AI, digital health, large language models, personalized healthcare, retrieval-augmented generation, clinical decision support, primary care, telemedicine

## Abstract

**Background/Objectives**: The demand for accessible and reliable digital health services has increased significantly in recent years, particularly in regions facing physician shortages. HomeDOCtor, a conversational AI platform developed in Slovenia, addresses this need with a nationally adapted architecture that combines retrieval-augmented generation (RAG) and a Redis-based vector database of curated medical guidelines. The objective of this study was to assess the performance and impact of HomeDOCtor in providing AI-powered healthcare assistance. **Methods**: HomeDOCtor is designed for human-centered communication and clinical relevance, supporting multilingual and multimedia citizen inputs while being available 24/7. It was tested using a set of 100 international clinical vignettes and 150 internal medicine exam questions from the University of Ljubljana to validate its clinical performance. **Results**: During its six-month nationwide deployment, HomeDOCtor received overwhelmingly positive user feedback with minimal criticism, and exceeded initial expectations, especially in light of widespread media narratives warning about the risks of AI. HomeDOCtor autonomously delivered localized, evidence-based guidance, including self-care instructions and referral suggestions, with average response times under three seconds. On international benchmarks, the system achieved ≥95% Top-1 diagnostic accuracy, comparable to leading medical AI platforms, and significantly outperformed stand-alone ChatGPT-4o in the national context (90.7% vs. 80.7%, *p* = 0.0135). **Conclusions**: Practically, HomeDOCtor eases the burden on healthcare professionals by providing citizens with 24/7 autonomous, personalized triage and self-care guidance for less complex medical issues, ensuring that these cases are self-managed efficiently. The system also identifies more serious cases that might otherwise be neglected, directing them to professionals for appropriate care. Theoretically, HomeDOCtor demonstrates that domain-specific, nationally adapted large language models can outperform general-purpose models. Methodologically, it offers a framework for integrating GDPR-compliant AI solutions in healthcare. These findings emphasize the value of localization in conversational AI and telemedicine solutions across diverse national contexts.

## 1. Introduction

Access to timely and reliable healthcare remains a persistent challenge in many countries, especially in areas with physician shortages or fragmented service delivery. In Slovenia, approximately 143,000 residents are currently without a personal physician, while more than 230,000 women lack access to a gynecologist. In neighboring North Macedonia, demographic disparities further complicate healthcare delivery, with low urbanization, an aging population, and systemic barriers for individuals with disabilities [1,2,3,4]. These realities point to an urgent need for scalable, intelligent systems that can extend high-quality medical guidance beyond traditional clinical infrastructure.

Recent advances in generative artificial intelligence (GenAI), particularly large language models (LLMs), have introduced new possibilities for digital health support. LLMs are increasingly being applied to triage, education, and decision-support tasks, showing high potential in controlled studies. For instance, two years ago, Ayers et al. [5] compared AI-generated responses to those of licensed physicians in online patient forums and found that the chatbot answers were rated significantly higher in both quality and empathy. Goh et al. [6] demonstrated that GPT-4 outperformed physicians using traditional resources in diagnostic tasks. However, their study also highlighted that the best outcomes occurred with AI systems, and in real life, when physicians effectively integrated LLMs into their diagnostic workflow, emphasizing the value of human-AI collaboration.

Despite the growing interest in generative AI applications in healthcare, most LLM-based tools are developed as general-purpose assistants and lack alignment with national clinical protocols, linguistic norms, and regulatory requirements. Although a few comparable systems exist, they are largely absent from the peer-reviewed scientific literature. For instance, in Germany, the Bertelsmann Stiftung has developed a digital assistant as part of its Digital Patient initiative, informally referred to as LEA, which aims to support users in navigating digital health services and understanding the DiGA framework. This situation underscores a critical research gap: the need to investigate how LLMs can be effectively adapted to and evaluated within specific national healthcare contexts. In particular, there is limited empirical evidence on the performance, safety, and utility of such systems when grounded in local medical guidelines and deployed under real-world conditions. To address this gap, the present study introduces and evaluates HomeDOCtor, a fully localized, regulation-compliant AI chatbot designed for the Slovenian healthcare system.

The need for deploying such tools is especially evident in underserved healthcare environments. Motivated by these challenges, the HomeDOCtor system was developed as a conversational AI platform first tailored to Slovenian medical protocols. Built on a modular architecture that integrates retrieval-augmented generation (RAG) with a Redis-based vector database of curated national guidelines, the system currently supports multilingual and multimedia inputs and operates 24/7. Based on our evaluations, HomeDOCtor outperforms conventional models from OpenAI due to its innovative design and methodology, which incorporates RAG. It is created to provide human-centered, protocol-consistent primary care support.

The study includes several of the most relevant LLMs available at the time of evaluation, encompassing both commercial and open-source models, such as GPT-4o (OpenAI), Gemini 2.5 (Google), and Gemma 3 (Meta via Ollama), to enable a comparative analysis across different levels of accessibility, licensing, and model architecture. These models were selected based on their availability, multilingual support, and demonstrated performance in previous clinical LLM studies. Importantly, the system is model-agnostic and can integrate any language model as needed. The HomeDOCtor variants incorporate these models through a modular orchestration framework, which enables domain-specific grounding via RAG.

The study aims to evaluate the diagnostic performance and practical viability of HomeDOCtor, a nationally adapted, large language model-based conversational AI system designed to support primary care guidance. This paper presents an assessment of HomeDOCtor based on international benchmark cases and national medical exam questions and examines the broader implications of local adaptation in AI-powered digital health systems. Specifically, it evaluates the system’s diagnostic accuracy, responsiveness, and contextual appropriateness using standardized international clinical vignettes and validated Slovenian medical school materials.

While generative AI is being increasingly explored in healthcare, few real-world studies have systematically evaluated the effectiveness of nationally tailored, regulation-compliant LLM systems that reflect local clinical protocols and linguistic norms. To the best of our knowledge, this is the first evaluation of a conversational AI system that fully integrates nationally curated medical knowledge into an RAG pipeline for live use within a European healthcare context. The original contribution of this work lies in demonstrating that such a localized approach can meaningfully outperform general-purpose models in country-specific applications, while remaining General Data Protection Regulation (GDPR)-compliant, clinically relevant, and publicly accessible.

The remainder of this paper is structured as follows: Section 2 reviews related work; Section 3 describes the system architecture and methodology; Section 4 presents the evaluation results; Section 5 discusses key implications and limitations; and Section 6 concludes with practical recommendations and directions for future research.

## 2. Related Work

Recent advances in generative AI have catalyzed the development of conversational tools in healthcare, although their real-world deployment remains limited due to technical, legal, and ethical challenges. Initial implementations typically focused on narrow use cases, such as insulin titration [7], medication adherence [8], pandemic response [9], and chronic disease monitoring [10]. These systems offered real-time, personalized advice but were constrained by domain specificity and limited adaptability.

With the emergence of large language models (LLMs), tools like ChatGPT are increasingly being used for backend decision support and patient-facing interactions. Fujita et al. [11] evaluated ChatGPT-4.0 in orthopedic postoperative care, while Liu et al. [12] reviewed its clinical potential more broadly. Arvidsson et al. [13] found GPT-4’s diagnostic performance on Swedish family medicine exams to be comparable to that of licensed physicians. Similarly, Ayoub et al. [14] reported that ChatGPT outperformed Google Search in clinical relevance and accuracy, highlighting the growing promise of LLMs in health information retrieval. Additional studies by Mehnen et al. [15] and Sebastian and Pragna [16] evaluated the diagnostic accuracy and ethical robustness of these systems in varied contexts.

Goodman et al. [17] further assessed ChatGPT’s reliability by evaluating its responses to 284 physician-generated questions across 17 specialties. While the system achieved high median scores for accuracy and completeness, it showed weaknesses in complex queries, especially those requiring clarification or clinical nuance. The study underscores the promise of LLMs, but also the need for robust safeguards against hallucinations and the absence of contextual prompts.

Alongside technical performance, researchers have emphasized the importance of transparency and accountability in clinical AI. Panigutti et al. [18] outlined how explainable AI (XAI) techniques, such as feature attribution and counterfactual reasoning, can enhance clinician trust. However, regulatory frameworks remain a bottleneck. The EU AI Act [19] classifies diagnostic systems as “high-risk”, while GDPR [20] imposes strict requirements for consent, data minimization, and retention. Some scholars warn that overlapping legal constraints could hinder innovation [21], though others highlight the role of regulation in ensuring fairness and safety [22,23].

Beyond text-only applications, pre-LLM chatbot research explored multimodal and hybrid AI approaches. Tjiptomongsoguno et al. [24] reviewed 27 early systems that combined natural language processing (NLP) with machine learning (ML) methods such as SVMs, Naïve Bayes, LSTMs, and seq2seq models. These chatbots supported use cases including mental health, diabetes monitoring, emergency triage, and remote vital sign tracking. However, limitations included noisy datasets, constrained linguistic flexibility, and reduced empathy in user interactions.

In a similar vein, Jegadeesan et al. [25] developed a symptom-driven diagnostic chatbot using K-nearest neighbors and rule-based logic. The system classified conditions as minor or serious, offering home remedies or referral guidance accordingly. Though achieving 82% accuracy and 24/7 usability, it lacked integration with formal clinical standards or real-world deployment.

Several newer systems aim to enhance flexibility and responsiveness using LLM-based architectures. Bertl [26] applied generative tools for hypertension management, while Bhatt and Vaghela [27] developed Med-Bot, an English-only AI assistant built on LLaMA-2, using LangChain and ChromaDB to process PDF medical literature. While technically advanced, its scope was limited by the language, lack of national protocol alignment, and absence of clinical evaluation.

Earlier frameworks like Doctor-Bot [28] and voice-based triage tools [29] also underscore the growing demand for accessible home-based AI systems. Yet, they typically rely on predefined flows or shallow reasoning, limiting adaptability and personalization.

Despite this growing body of work, few systems incorporate national medical protocols, nor do they address regulatory alignment, linguistic specificity, and live deployment in a unified framework. To the best of our knowledge, no existing chatbot published in peer-reviewed literature fully localizes its retrieval and generation pipelines to country-specific healthcare needs.

In response, HomeDOCtor implements a nationally adapted architecture built on a retrieval-augmented generation (RAG) framework enriched with curated Slovenian medical content. Unlike generic LLMs, it generates context-aware, protocol-compliant, and linguistically relevant responses while maintaining GDPR compliance. This localized approach improves diagnostic accuracy and enhances user trust, offering a scalable and replicable model for the development of personalized AI-driven healthcare systems in national contexts.

## 3. Materials and Methods

### 3.1. System Development and Architecture

The HomeDOCtor platform was developed and started as an independent project and was later upgraded within the European Horizon-funded ChatMED project (Grant Agreement 101159214), which aims to support the co-development of national conversational AI systems tailored for health guidance. The broader objective of the project is to improve access to multilingual and trustworthy digital health services across diverse European healthcare environments through localized knowledge integration, user-centered design, and regulatory alignment. While the system has been deployed in Slovenia, additional adaptations for other national contexts are currently underway.

The underlying knowledge structure in HomeDOCtor builds upon the Insieme ontology (https://ise-emh.eu/), a curated resource originally created in the Interreg Italia–Slovenia project. This ontology represents key concepts in Slovenian family and internal medicine, such as symptoms, diagnoses, clinical actions, and referral triggers, and is combined with guideline-based content from several authoritative national sources.

To enable RAG, the HomeDOCtor architecture uses a modular pipeline with a Redis-based semantic vector database at its core (see Figure 1). Medical documents are embedded into this vector store using OpenAI’s text-embedding-3-large model, which was chosen for its superior multilingual retrieval performance and compatibility with downstream LLMs. This model encodes curated Slovenian-language clinical content into high-dimensional vector representations suitable for semantic search.

The medical knowledge used in the system was curated from the following:The Slovenian Manual of Family Medicine;Public treatment protocols and pathways issued by Slovenian health authorities;Official discharge instructions and health promotion materials;Insieme ontology knowledge graphs and formalized rulesets.

These documents were preprocessed manually to ensure consistency, remove irrelevant formatting, and split content into semantically meaningful chunks. Each chunk typically represents a discrete unit of clinical logic, such as the following:A symptom-diagnosis decision rule,A triage or escalation recommendation,A medication contraindication warning,A structured care pathway (e.g., for fever, back pain, chest discomfort).

The text fragments were transformed into vector embeddings and stored in a Dockerized Redis Stack instance, where they were indexed with metadata (e.g., document source, medical domain, language, and content type). This enables targeted semantic search across national medical content when a user query is submitted.

HomeDOCtor was designed from the outset to comply with the GDPR and related Slovenian data protection laws, especially those concerning sensitive health data. To meet these requirements, (i) no user data are stored beyond the active session; (ii) all interactions are stateless, no profiles or longitudinal records are maintained; (iii) inputs are processed in-session and deleted when the session ends; (iv) all prompts and responses remain anonymous and unlinked to identifiable users.

While this design ensures maximal legal compliance and user privacy, it imposes several usability trade-offs: (i) users must re-enter personal health information each time they use the system; (ii) the system cannot remember previous conversations or offer longitudinal guidance; (iii) features like personalization, health history tracking, or follow-up alerts are currently unavailable.

These limitations were implemented to avoid requiring complex consent flows and to eliminate legal risks associated with retaining or linking health data. In practice, internal testing showed that session-based interaction was sufficient for ad-hoc self-care, triage, and diagnostic guidance, especially in scenarios where the user is seeking quick, anonymous support. Future integration with national health portals may reintroduce personalization under informed consent workflows.

HomeDOCtor is composed of several modular and interoperable components:**Flutter Frontend UI**: A cross-platform, multilingual interface where users can input queries (text, image, or document), choose the language and model, and view structured outputs (e.g., diagnostic replies, care instructions, cited medical sources) (Flutter version 3.24.5).**FastAPI Backend**: A Python-based backend that manages request routing, session tracking, and communication between frontend and backend modules. It orchestrates prompt assembly and LLM calls based on user input (Python version 3.12).**Redis Database (Redis Stack)**: (i) Semantic vector store for embedded Slovenian medical texts; (ii) temporary session memory to maintain conversational context during a single interaction; (iii) queue and feedback management for API access and future system refinement; (iv) integration with a symbolic knowledge graph for hybrid reasoning; (v) prompt Engineering and RAG Orchestration: receives input and dynamically builds prompts. It retrieves the top 3 to 5 most semantically relevant text snippets from the Redis vector database based on cosine similarity. These snippets are integrated into the prompt using a structured template that includes the following (Redis database stack 7.2.0):A system instruction block (ensuring safe, guideline-based replies),Retrieved medical content (clearly separated and marked),The user’s original query.**LLM Orchestration:** Interfaces with selected language models. The system currently supports models including GPT-4o (OpenAI), Gemini 2.5 (Google), and Gemma 3 (Meta via Ollama) through OpenRouter or local inference. These models were selected based on multilingual capability, licensing variety, and clinical benchmark performance. The architecture is model-agnostic and prepared for integration with the Model Context Protocol (MCP) to streamline future LLM workflows and enable fully local inference in privacy-sensitive contexts. The different HomeDOCtor variants (e.g., 4o, o3 mini high, Gemini 2.5) are identical in terms of architecture, RAG pipeline, and the curated national medical corpus. The primary distinction between versions lies in the underlying language model used for generation. No additional model-specific fine-tuning was applied; instead, each variant inherits the same prompt template, semantic retrieval logic, and medical grounding. This allows for consistent comparison of diagnostic performance across LLMs while controlling for all other components of the system pipeline.

### 3.2. End-to-End Operational Flow

The following sequence outlines the complete lifecycle of a user interaction with the HomeDOCtor platform, from initial query submission to the final response. The architecture is designed to ensure privacy, contextual relevance, and guideline compliance while operating under GDPR constraints. Each step is modular and supports secure, real-time processing of multilingual and multimodal medical queries:(1)**User Submission**: Through the Flutter frontend, the user sends a text query (optionally with images or documents) and may choose a language and LLM model. In compliance with GDPR requirements, no personal data are stored beyond the session, and all submissions remain anonymous.(2)**Request Reception**: FastAPI receives the request, validates input, identifies the user’s session, and begins contextual processing. Due to the legal requirement for statelessness, each interaction is handled independently, without persistent user history.(3)**Session Retrieval**: The backend queries Redis to fetch session data, manage queue status, and restore dialogue context within the current session only. This ensures session continuity without long-term data retention.(4)**Initial Prompt Construction**: The prompt engineering module receives the input and metadata, pre-processes any multimodal elements, and builds the base prompt. At this stage, usability is balanced with legal constraints—e.g., users must re-enter relevant details each time, which avoids consent management but may reduce personalization.(5)**RAG Preparation**: Within prompt engineering, the RAG orchestrator determines if external knowledge is needed to enrich the prompt.(6)**Semantic Query**: The orchestrator issues a semantic search to Redis’s vector store to find pertinent medical document fragments.(7)**Prompt Augmentation**: Redis returns relevant chunks, which the orchestrator appends to the base prompt, creating a knowledge-grounded version. Typically, three to five snippets are retrieved based on cosine similarity. These represent the most semantically relevant content from curated Slovenian clinical guidelines. The structured prompt includes: (1) a system instruction header for safety and guideline compliance; (2) retrieved snippets, clearly marked and separated; and (3) the user’s original query. This process ensures the model receives a contextually grounded input that enhances reliability and medical relevance.(8)**Model Selection**: The enriched prompt is sent to LLM orchestration, which chooses the appropriate model.(9)**LLM API Call**: The orchestrator calls the selected LLM (e.g., OpenAI, a local Ollama model, or via OpenRouter) to generate a response.(10)**Response Handling**: The LLM’s output is returned to the response handler, where it is post-processed for user readability and safety compliance.(11)**Multimedia Fetch** (if needed): If additional media is required, the conversation logic invokes the multimedia module to perform web searches (via LLM or Google PSE) and gather images, links, or videos.(12)**Final Assembly:** The response handler merges the LLM text with any multimedia, formats it, and routes the complete package back through FastAPI.(13)**User Display:** The Flutter frontend receives and displays the final output in the form of text, cited sources, and multimedia, offering a context-aware, interactive, yet fully GDPR-compliant experience.

The system was deployed nationally across Slovenia at the end of 2024. It was integrated with the Insieme electronic health infrastructure and evaluated in both desktop and mobile settings. The target users were laypersons seeking internal medicine guidance, especially in regions with limited physician availability [1]. The deployment supported five interface languages and enabled users to receive structured medical outputs, self-care guidance, and doctor-referral suggestions. The prompt consists of 250 lines of structured instructions designed to optimize user experience (UX) for patients, particularly those with psychological or neurocognitive conditions. These instructions employ patient-centered communication strategies, incorporating plain language principles and empathetic phrasing to enhance comprehension, reduce anxiety, and promote adherence.

### 3.3. Evaluation Strategy

We evaluated HomeDOCtor’s diagnostic accuracy and relevance through two experiments:

International Benchmarking: A standardized set of 100 clinical vignettes from the Avey AI Benchmark Suite [30] was used to compare HomeDOCtor against several leading LLMs. All models were evaluated using the same one-shot prompting method [6].

National Contextualization: A separate test set of 150 validated internal medicine multiple-choice questions was drawn from Slovenian medical school materials. Responses were assessed against official answers and reviewed by medical professionals and senior students [10]. All questions were administered in Slovenian, using formal phrasing as found in university materials. Responses generated by the models were independently reviewed and scored by two final-year medical students and cross-verified by a senior clinician. Ambiguous cases were resolved by consensus. The evaluation process was blinded to model identity to prevent bias.

Each model response was independently evaluated by final-year medical students using a predefined scoring rubric, with results compared against the official answer. In cases where the evaluators disagreed, discrepancies were discussed and resolved through consensus. When consensus was not immediately reached, a senior clinician served as the final arbiter. Although this process ensured consistency in judgment, no formal inter-rater reliability metric (e.g., Cohen’s kappa) was calculated in this study. We acknowledge this as a limitation and plan to incorporate quantitative agreement analysis in future evaluations.

In addition, the remarks obtained by the users were studied, and modifications were implemented to improve the system accordingly, if possible.

Performance metrics included Top-1 diagnostic accuracy, latency (response time), contextual appropriateness, and safety-sensitive behaviors (e.g., escalation to physician recommendation). Statistical comparisons between models were performed using pairwise two-proportion z-tests with Bonferroni correction [23].

### 3.4. Safety, Privacy, and Usability Considerations

HomeDOCtor was developed with strict attention to patient safety and data protection. All user interactions are session-based and anonymized, with no personal identifiers stored. The platform complies with GDPR requirements for data minimization, consent, and storage duration [19]. In addition, HomeDOCtor is still treated as a research prototype of the European project ChatMED.

Due to national legal requirements that mandate all health-related data remain within Slovenia, and the fact that the best-performing LLMs are often hosted in the United States, HomeDOCtor does not retain user data beyond the session. Instead, all personal information is entered by the user at the time of interaction and is discarded at the end of the session. Consequently, users must either re-enter relevant personal health information each time or retrieve it from linked national health infrastructure services.

Usability was evaluated informally during development via a small group of medical professionals and end-users who willingly provided structured feedback on clarity, accessibility, and perceived safety. The interface was refined through iterative co-design based on accessibility principles [4], including options for simplified language, dark mode, and keyboard-only navigation. A formal usability study is planned for the next release phase.

## 4. Experimental Results

To evaluate the effectiveness of the HomeDOCtor platform, we conducted two structured experiments focusing on diagnostic accuracy, contextual relevance, and performance in both international and national healthcare scenarios.

### 4.1. Results for Clinical Vignettes

We benchmarked HomeDOCtor and several other LLMs using a standardized set of 100 clinical vignettes from the Avey AI Benchmark Suite [30]. These cases encompassed a range of difficulties and were independently verified by medical experts for realism and completeness. Each model received the same one-shot prompt (including basic clinical metadata) and generated a Top-1 diagnosis without follow-up clarification.

As shown in Table 1, HomeDOCtor 4o and HomeDOCtor o3 mini high achieved the highest performance (99/100 correct diagnoses), followed closely by ChatGPT-4o and other commercial models. Even smaller, open-source models demonstrated strong performance, indicating the practical viability of such models in clinical support tools.

These findings suggest that HomeDOCtor’s architecture, despite being focused on localized corpora, achieves competitive diagnostic accuracy across general medical cases.

Complementary internal evaluations on typical real-world cases up to May 2025, representative of those commonly encountered by general practitioners, revealed comparable trends. All top-performing systems consistently achieved near-perfect diagnostic accuracy. Even relatively lightweight, open-source models such as *Gemma 3* demonstrated strong performance, reaching approximately 95% accuracy. These findings reinforce the practical feasibility of deploying cost-effective LLMs in clinical settings. Given the uniformly high performance, differences between top models were not statistically significant for this subset. Given the uniformly high accuracy across models, no statistically significant differences were observed in this subset of cases.

### 4.2. Results in National Medical Exam Context

To evaluate model performance in a nationally relevant context, a dataset of 150 medical examination questions was compiled from study materials available to students at the Faculty of Medicine, University of Ljubljana. The questions primarily covered internal medicine, with additional items from primary care and surgery, all aligned with internal medicine themes. Responses were blinded and independently evaluated by medical students and experts.

The models were evaluated using a one-shot prompting strategy, in which each question was presented to the systems in a standardized, identical format to ensure fairness and eliminate prompt engineering bias. The comparison included HomeDOCtor variants based on LLMs, alongside leading OpenAI models (e.g., ChatGPT) and several other LLMs. All models received the same set of 150 questions without follow-up clarification, replicating real-world diagnostic conditions where user input may be brief or ambiguous. All HomeDOCtor variants used the same prompt structure, retrieval method, and medical knowledge base, ensuring that the only variable across models was the underlying LLM engine. This setup allows for an isolated assessment of each language model’s diagnostic reasoning capacity within a consistent system framework.

Figure 2 presents a vertical bar chart showing the total number of correct responses (out of 150) for each evaluated model. Among the systems tested, ChatGPT o3 mini high achieved the highest raw accuracy, while Gemma 3 exhibited the lowest performance. The results illustrate a performance gradient between cutting-edge commercial models and smaller, open-source alternatives.

Figure 3 provides a focused comparison between HomeDOCtor variants and the general-purpose LLMs. It highlights the impact of fine-tuning and domain-specific adaptations on diagnostic accuracy. Across all variants, HomeDOCtor models consistently outperformed or matched their baseline counterparts, demonstrating the value of incorporating national medical guidelines and localized knowledge into model workflows when dealing with national cases and applying local protocols. This chart reinforces the effectiveness of targeted LLM customization for healthcare applications.

### 4.3. Statistical Comparison for National Tests

To assess whether differences in diagnostic performance between models were statistically significant, we conducted three pairwise two-proportion z-tests (see Table 2). Each comparison was based on 150 independent questions from the national evaluation dataset. The comparisons included the following:−*HomeDOCtor* 4o vs. ChatGPT-4o−*HomeDOCtor* o3 mini high vs. ChatGPT o3 mini high−*HomeDOCtor* Gemini 2.5 vs. Google Gemini 2.0.

The two-proportion z-test is appropriate for binary outcome data (correct = 1, incorrect = 0) and tests whether observed differences in success rates between two models could plausibly occur by chance. To control for multiple comparisons and maintain an overall Type I error rate of 5%, we applied a Bonferroni correction, adjusting the significance threshold to α_a_*d**j* = 0.05/3 ≈ 0.0167.

Results of the Pairwise Comparisons:−*HomeDOCtor* 4o vs. ChatGPT-4o: *HomeDOCtor* 4o achieved 136 correct answers (90.7%), compared to ChatGPT-4o’s 121 correct (80.7%). The resulting z-statistic was 2.47 with a *p*-value of 0.0135. Since *p* < 0.0167, this difference is statistically significant.−HomeDOCtor o3 mini high vs. ChatGPT o3 mini high: *HomeDOCtor* scored 140/150 (93.3%), while ChatGPT o3 mini high scored 131/150 (87.3%). The z-value was 1.76 with a *p*-value of 0.0789, which does not meet the Bonferroni-adjusted threshold for significance.−HomeDOCtor Gemini 2.5 vs. Google Gemini 2.0: The scores were 116/150 (77.3%) and 107/150 (71.3%), respectively, yielding a z-value of 1.18 and *p* = 0.2352. This difference was not statistically significant.

Only the comparison between *HomeDOCtor* 4o and ChatGPT-4o showed a statistically significant difference after Bonferroni correction. These findings suggest that HomeDOCtor’s RAG architecture, supported by a Redis-powered vector store, can provide a measurable accuracy advantage over standalone LLMs. However, not all version-to-version improvements reached statistical significance, indicating that gains may be incremental or context-dependent.

For reference, the two-proportion z-test is defined as follows:(1)$$P=\frac{True}{All}\cdot 100\%$$(2)$$N=\left(100-P\right)$$(3)$$\hat{p}=\frac{x_{1}+x_{2}}{n_{1}+n_{2}}$$(4)$$SE=\sqrt{\left(\hat{p}\left(1-\hat{p}\right)\left(\frac{1}{n_{1}}+\frac{1}{n_{2}}\right)\right)}$$(5)$$z=\frac{\frac{x_{1}}{n_{1}}-\frac{x_{2}}{n_{2}}}{SE}$$(6)$$p=2\cdot \left[1-\Phi \left(\left|z\right|\right)\right],$$
where Φ is a standard normal cumulative distribution function.

Since the three tests are independent of each other, the Bonferroni correction was used: $\alpha_{n}=\frac{\alpha }{m}$, where m = 3.

## 5. Discussion

The results from both international and national evaluations demonstrate that HomeDOCtor provides competitive diagnostic accuracy while delivering locally contextualized, guideline-compliant responses. The system’s strong performance on the Avey AI international vignette suite confirms its ability to generalize across diverse clinical scenarios, matching or exceeding outputs from state-of-the-art commercial models (e.g., ChatGPT-4o and Gemini) [30]. This reflects similar trends found in prior benchmarking studies, where LLMs have achieved diagnostic parity or superiority over traditional clinical tools [6,13,14,15].

However, the most noteworthy finding lies in HomeDOCtor’s performance on nationally contextualized medical exam questions. In this setting, the platform significantly outperformed baseline LLMs, particularly in the case of HomeDOCtor 4o vs. ChatGPT-4o (*p* = 0.0135). These findings underscore the value of tailoring AI systems to local medical protocols and linguistic contexts—a recurring theme in current AI-in-healthcare research [24,29], especially in regions with language-specific health protocols and limited digital health infrastructure. Localized RAG, backed by curated national guidelines, appears to offer tangible gains in clinical alignment and response reliability.

While the HomeDOCtor 4o variant demonstrated a statistically significant improvement over ChatGPT-4o in national exam performance, the differences between other models (e.g., HomeDOCtor o3 mini high vs. ChatGPT o3 mini high) did not reach statistical significance. This lack of statistical significance should not be interpreted as evidence of equal performance, but rather as an indication that differences may be context-dependent or sample-size limited. It also reflects the narrowing performance margins among high-performing LLMs, where even small improvements require larger sample sizes or more targeted evaluation frameworks to detect reliably. In our case, performance clustering near 90% suggests that while national adaptation improves alignment with Slovenian protocols, the observable advantage may vary depending on question phrasing, complexity, or clinical specificity. While our evaluation focused primarily on accuracy metrics, we acknowledge the absence of a formal error analysis as a limitation. Informal internal reviews of incorrect responses revealed that most errors occurred in cases where symptoms overlapped across multiple conditions (e.g., chest discomfort, fever with rash) or where user input was minimal or vague. Importantly, none of the reviewed outputs included harmful, contraindicated, or unsafe recommendations. Most misclassifications resulted in overly cautious triage (e.g., suggesting a doctor visit for self-limiting conditions), reflecting the system’s safety-first design. Nonetheless, a detailed categorization of errors, distinguishing between benign misclassifications, missed escalations, and potential risks, will be a focus of future studies. This will allow us to better refine prompt structures and retrieval logic to minimize failure modes in edge-case scenarios.

These results align with the special issue’s emphasis on leveraging digital innovation to achieve equitable, personalized healthcare delivery. HomeDOCtor shows how generative AI systems, when carefully adapted, can act as accessible, low-barrier digital triage tools that support self-care, improve patient empowerment, and reduce unnecessary clinical visits. Particularly for populations historically marginalized by digital health trends, such as elderly users, people with disabilities, or those in rural settings, HomeDOCtor’s multilingual interface, following cultural habits and local protocol grounding, promotes trust and engagement [1,2,3,4,23].

From a systems design perspective, this study reinforces the importance of integrating domain-specific corpora, supported by domain-specific validation and iterative co-design processes tailored to clinical contexts. Mobile-phone-based applications in clinical monitoring and risk detection, using accelerometer sensors [31], demonstrate the potential for multimodal extensions to enhance personalization. Similarly, HomeDOCtor can use local sensors in the mobile phone of a user. Experience from European health technology initiatives underscores the importance of interoperability and cross-border adaptation—especially in aging and active living domains [29]. While generic LLMs often struggle with regional compliance or nuance, HomeDOCtor’s modular orchestration and constrained prompting workflows address these issues directly. This aligns with best practices identified in studies on human–AI collaboration and explainable AI (XAI) in clinical settings [17,18,24].

Nonetheless, the platform has limitations. It remains a research prototype and is not yet integrated with patient electronic health records (EHRs) or biometric sensors, although such features were developed but not deployed due to GDPR issues. Usability testing was limited to informal expert feedback; a formal usability study, as recommended in other telehealth evaluation work [4,8], is planned for a future release with written user consent. Also, the focus on internal medicine limits generalizability to other domains such as pediatrics, elderly care, or mental health, which would require separate guideline integration.

While the system demonstrates successful performance, several important considerations remain. First, the development and deployment process involved close consultation with legal experts to ensure full compliance with GDPR and related data protection frameworks. To avoid complications with data storage and consent, all personal information remains on the user’s side and is discarded after the session, effectively shifting responsibility and legal risk away from the platform provider. Second, concerns about incorrect recommendations are valid; however, this is a challenge shared by all clinical decision systems, including human practitioners. Numerous studies suggest that AI-based systems may already outperform average physicians in consistency and recall [6,15], and at the same time, HomeDOCtor is explicitly positioned as a support tool providing advice, and not a replacement for human judgment. Third, high-quality service depends not only on algorithmic performance but also on the user’s interaction time and engagement. While many users engage for only 1–2 min, richer and more personalized recommendations emerge in sessions lasting 10–15 min, particularly when users provide comprehensive symptom descriptions or relevant medical history, enabling more accurate and tailored recommendations. Fourth, the system’s overall quality is consistently rated as excellent in internal evaluations and field use. The system generates timely, accurate, and context-sensitive responses, typically within a couple of seconds, adapting to user needs in both Slovenian and other supported languages. Finally, the system is designed to maintain a high standard of user experience. Unlike some human and artificial interfaces (surprisingly, e.g., communicating with patients as “friends”), it never becomes impatient, dismissive, or impolite. HomeDOCtor always communicates respectfully, using appropriate formal address in Slovenian and other context-aware language settings, further reinforcing user trust and comfort.

Expanding to other jurisdictions will require collaboration with international health authorities to adapt clinical knowledge bases, adjust conversational flows for cultural relevance, and ensure compliance with varying regulatory regimes (e.g., HIPAA in the U.S., EU Medical Device Regulation), guided by lessons from prior comparisons of EU healthcare platforms [32]. As generative AI in healthcare falls under overlapping regulatory frameworks, such as the EU AI Act’s designation of diagnostic support systems as high-risk, requiring transparency, human oversight, and post-market surveillance [17,18,19], HomeDOCtor meets these standards through detailed logging of AI decision pathways, routine model output audits, and user-friendly explanations of recommendation logic. Ethical principles concerning informed consent, data minimization, and equitable access are incorporated via a human-centered co-design process involving patients, clinicians, and disability advocates [33].

Overall, the findings confirm the feasibility of a nationally adapted LLM-based conversational AI system that delivers medically useful, trusted, and responsive digital guidance. With careful design, domain grounding, and regulatory alignment, such systems can meaningfully augment primary care capacity—especially for users with special needs or in underserved or linguistically isolated populations—and align with broader European strategies for inclusive, equitable digital health transformation [33].

## 6. Conclusions

This study demonstrates the practical potential of nationally adapted, LLM-powered conversational systems to contribute to the digital transformation of healthcare. Designed with inclusivity, legal compliance, and clinical relevance in mind, HomeDOCtor has proven to be a reliable and user-aligned digital health tool. By combining an advanced RAG, curated national guidelines, and multilingual interaction, the system delivers trustworthy, protocol-compliant recommendations while respecting linguistic and cultural norms, including formal address in proper communication styles and simplified, accessible interfaces for advanced and digitally marginalized users [3].

Theoretically, this work highlights the importance of national adaptation in making AI systems clinically relevant and trustworthy at the point of care. It suggests that domain-specific retrieval and localized prompt engineering can improve the contextual grounding of LLMs, leading to higher diagnostic accuracy in national settings. Methodologically, the system offers a replicable architecture for GDPR-compliant AI deployment in healthcare, balancing privacy with usability through a stateless session-based design and modular integration of multilingual LLMs. Technically, the use of a Redis-based semantic vector database and RAG orchestration demonstrates a scalable approach to structured knowledge retrieval for safe and context-aware generation.

From a practical and managerial perspective, we recommend that public health stakeholders and decision-makers consider AI-based conversational agents like HomeDOCtor as complementary tools for managing demand in primary care, particularly in underserved or demographically vulnerable regions. Policy frameworks should encourage localization and regulation-aware design from the outset. Institutions deploying such systems should ensure the inclusion of clinicians, patients, and legal experts throughout the development lifecycle. Additionally, national healthcare authorities may benefit from co-developing curated, machine-readable clinical guidelines to support safe and accurate AI-driven guidance.

This study has several limitations. First, the system was evaluated primarily in the domain of internal medicine; future expansions should cover pediatrics, mental health, and geriatric care. Second, the usability evaluation was informal and limited to expert feedback; a formal user study is planned for the next release phase. Third, due to GDPR constraints, personalization features and long-term interaction histories are currently unavailable. Future versions may support these under consent-based workflows via national health portals. Lastly, while national deployment was successful, broader international applicability will require additional adaptation to local clinical protocols, regulatory frameworks, and language norms.

In summary, HomeDOCtor illustrates that large language model systems, when responsibly adapted to local healthcare contexts, can augment healthcare access, deliver high-quality guidance, and contribute to more equitable, personalized digital health systems. This approach aligns with broader European strategies for inclusive health innovation and offers a scalable foundation for AI-enhanced primary care.

## Figures and Tables

**Figure 1 healthcare-13-01843-f001:**
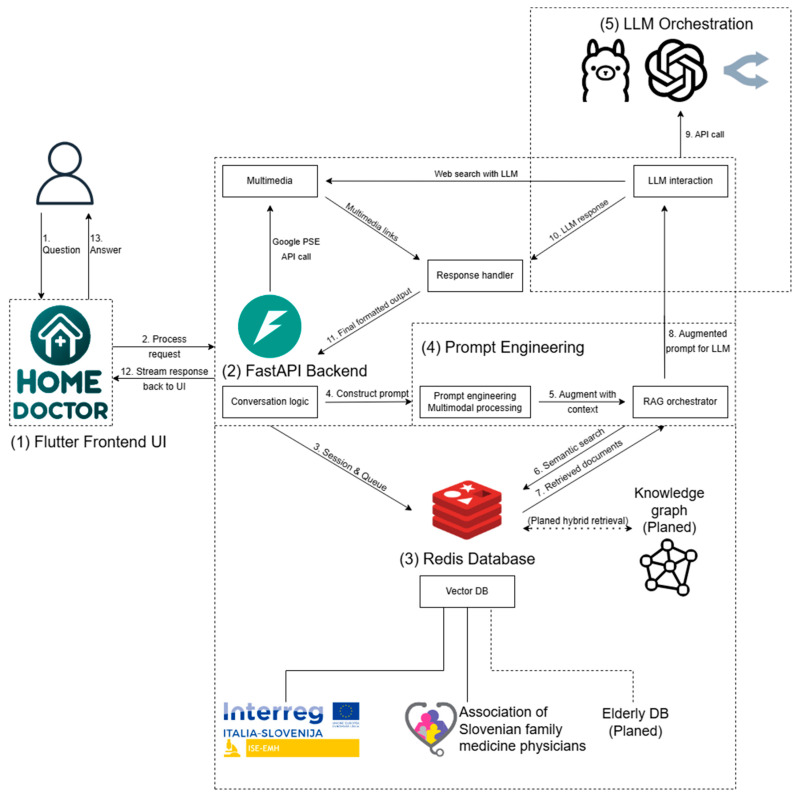
*HomeDOCtor* using RAG technique with additional data in the Redis vector database.

**Figure 2 healthcare-13-01843-f002:**
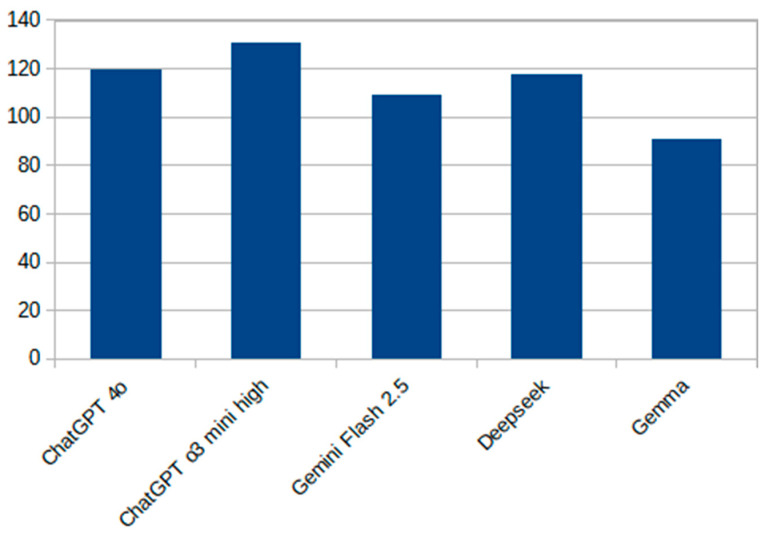
Comparison of large language models on a 150-question national dataset.

**Figure 3 healthcare-13-01843-f003:**
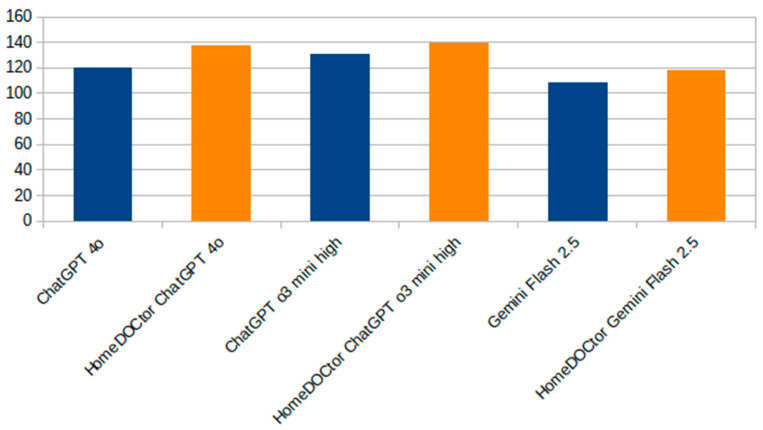
Comparison of HomeDOCtor variants and available large language models on a 150-question dataset.

**Table 1 healthcare-13-01843-t001:** Accuracy of *HomeDOCtor* variants and other LLMs: top 1, (N = 100 Questions).

Model	Top 1 Answer
*HomeDOCtor* 4o	99/100
*HomeDOCtor* o3 mini high	99/100
*HomeDOCtor* Gemini 2.5 Flash	98/100
ChatGPT-4o	98/100
ChatGPT o3 mini high	97/100
Gemini	96/100
Deepseek	95/100
Gemma 3	95/100

**Table 2 healthcare-13-01843-t002:** Pair comparison between models.

Comparison Between	Model 1 Rate	Model 2 Rate	Z-Statistic	*p*-Value	Significant
*HomeDOCtor* 4o vs. ChatGPT-4o	$\frac{136}{150}$	$\frac{121}{150}$	2.47	0.0135	Yes
*HomeDOCtor* o3 mini high vs. ChatGPT o3 mini high	$\frac{140}{150}$	$\frac{131}{150}$	1.76	0.0789	No
*HomeDOCtor* Gemini 2.5 vs. Google Gemini 2.0	$\frac{116}{150}$	$\frac{107}{150}$	1.18	0.2352	No

## Data Availability

The application is online and can be accessed: https://home-doctor.ijs.si/ (accessed on 10 June 2025). Note: HomeDOCtor can only be accessed through the national system to prevent overload caused by external use.

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
