# Peer review of "Evaluating a Nationally Localized AI Chatbot for Personalized Primary Care Guidance: Insights from the HomeDOCtor Deployment in Slovenia"

_healthcare, 2025, doi:10.3390/healthcare13151843_

Round 1
Reviewer 1 Report
Comments and Suggestions for Authors
Although this article is interesting, it needs to be examined in detail from many aspects and the following deficiencies need to be addressed.
1-The motivation for the study must be stated. Authors must do this by clearly stating the research gap.
2-It remains unclear which LLM models were used in the study. It should also be stated why the LLM model was chosen. This section should be clarified.
3-Which model or encoder is chosen for domain-specific embeddings?
4-Which model or encoder is chosen for domain-specific embeddings?
5-How are user questions semantically parsed during the query? Please explain.
6-How many snippets are collected? How are these snippets integrated into the prompt? Please provide detailed information.
7-How many snippets are collected? How are these snippets integrated into the prompt? Please provide detailed information.
8-Because the study is primarily focused on Internal Medicine, it cannot offer recommendations for other specialties. I suggest addressing this limitation in the discussion section.
Author Response
We appreciate the reviewer’s helpful observation, which has significantly strengthened the presentation of our study.
1-The motivation for the study must be stated. Authors must do this by clearly stating the research gap.
In response, we have revised the introduction to explicitly highlight the research gap motivating our work:
Despite the growing interest in generative AI applications in healthcare, most large language model (LLM)-based tools are developed as general-purpose assistants and lack alignment with national clinical protocols, linguistic norms, and regulatory requirements. Although a few comparable systems exist, they are largely absent from the peer-reviewed scientific literature. For instance, in Germany, the Bertelsmann Stiftung has developed a digital assistant as part of its Digital Patient initiative, informally referred to as LEA, which aims to support users in navigating digital health services and understanding the DiGA framework. This situation underscores a critical research gap: the need to investigate how LLMs can be effectively adapted to and evaluated within specific national healthcare contexts. In particular, there is limited empirical evidence on the performance, safety, and utility of such systems when grounded in local medical guidelines and deployed under real-world conditions. To address this gap, the present study introduces and evaluates HomeDOCtor, a fully localized, regulation-compliant AI chatbot designed for the Slovenian healthcare system.
2-It remains unclear which LLM models were used in the study. It should also be stated why the LLM model was chosen. This section should be clarified.
We thank the reviewer for this valuable observation. In the revised manuscript, we include: “The study includes several of the most relevant large language models (LLMs) available at the time of evaluation, encompassing both commercial and open-source models, such as GPT-4o (OpenAI), Gemini 2.5 (Google), and Gemma 3 (Meta via Ollama), to enable a comparative analysis across different levels of accessibility, licensing, and model architecture. These models were selected based on their availability, multilingual support, and demonstrated performance in previous clinical LLM studies. Importantly, the system is model-agnostic and can integrate any language model as needed. The HomeDOCtor variants incorporate these models through a modular orchestration framework, which enables domain-specific grounding via retrieval-augmented generation (RAG).”
3,4 -Which model or encoder is chosen for domain-specific embeddings?
We thank the reviewer for this important question. In the revised manuscript, we now clarify that the encoder used for generating all domain-specific vector embeddings was OpenAI’s text-embedding-3-large model, which at the time of deployment represented one of the most capable and widely used options for semantic search and dense retrieval tasks. This model was selected due to its strong performance on biomedical and multilingual benchmarks, as well as its compatibility with downstream LLMs used in the system. It encodes curated Slovenian-language medical documents, such as national clinical guidelines, internal medicine manuals, and Insieme ontology resources into high-dimensional vectors stored in the Redis semantic vector database. These embeddings serve as the backbone of the RAG pipeline, allowing the system to retrieve clinically relevant content based on semantic similarity rather than keyword matching.
5-How are user questions semantically parsed during the query? Please explain.
We appreciate this remark. In response, we have expanded the manuscript to include a detailed description of the full end-to-end operational flow of the system. This explanation has been added to the Materials and Methods section and outlines each processing step from user input to semantic parsing, prompt construction, retrieval, and LLM response generation, providing greater transparency into how the system handles user queries.
6-How many snippets are collected? How are these snippets integrated into the prompt? Please provide detailed information.
We thank you for this relevant question. In the revised manuscript, we have added further detail in the Prompt Engineering & RAG Orchestration section to clarify how semantic retrieval and prompt construction are performed. Specifically, the system retrieves the top 3 to 5 most semantically relevant text fragments (“snippets”) from the Redis-based vector database in response to the user’s input. The number of snippets is dynamically determined based on relevance thresholds and available prompt token space.
These retrieved snippets are directly integrated into the prompt using a structured template that includes:
- a system-level instruction block (guiding the LLM to provide guideline-compliant, empathetic medical advice),
- the retrieved contextual snippets (clearly marked and separated to avoid confusion),
- and the user’s original query.
This structure ensures that the LLM has direct access to relevant, localized medical knowledge while preserving clarity and coherence in the generated response. We have now included this explanation explicitly in the updated Materials and Methods section.
7,8 -Because the study is primarily focused on Internal Medicine, it cannot offer recommendations for other specialties. I suggest addressing this limitation in the discussion section.
We acknowledge this remark. In response, we have revised the Discussion section to explicitly acknowledge this limitation. As now stated:
“Additionally, the focus on internal medicine limits generalizability to other domains such as pediatrics, elderly care, or mental health, which would require separate guideline integration.”
This clarification helps set realistic expectations about the system’s current scope and paves the way for future domain-specific extensions. Thank you again for this helpful suggestion.
Reviewer 2 Report
Comments and Suggestions for Authors
Abstract
The abstract provides an overview of the study, but it needs to much more clearly and detailedly highlight the practical and theoretical implications of the paper. A separate section or a longer sentence dedicated to the implications would significantly enhance the usefulness of the abstract and offer the reader a more complete picture of the study's contribution.
Introduction
The introduction section establishes the context, but does not sufficiently articulate the precise purpose and objectives of the research, without transforming the general description of the problem into clear and measurable statements of purpose. It is desirable to clearly highlight the gaps in the existing literature that the current study intends to address.
Furthermore, the degree of novelty, although suggested by the description of the nationally adapted system, is not explicitly indicated through a statement of original contribution. Additionally, the structure of the article should be recommended, explaining what each subsequent section will cover.
Literature Review (Related Work)
Although the section includes sources relevant to the topic addressed, its structure can be improved through a more coherent presentation of the current state of knowledge in the field. A more in-depth analysis of the literature would allow for a solid understanding of the conceptual basis of the HomeDOCtor system. It is desirable for the article to analyze in more detail the connection between HomeDOCtor and other similar systems/applications/AI assistants mentioned, justifying the relevance of national adaptation.
Also, the section is relatively short, which may suggest that it does not cover in depth all the relevant literature necessary to fully support the contributions. Expanding this section would provide a more solid scientific foundation and increase the academic credibility of the work.
Materials and Methods
To consolidate the methodological section of the article, the authors should focus on clarifying technical details and presenting their evaluation approach more transparently. There is a need to discuss in more detail the trade-offs between GDPR compliance and user experience, specifically providing a transparent explanation of how data non-retention affects the usability of the system currently and why this approach was adopted for legal reasons.
Furthermore, it is advisable to provide more technical details about the architecture and data used by describing more precisely how the Slovenian medical guidelines are structured and integrated into the Redis vector database.
Discussions
Although the discussion section interprets the results, it is advisable to analyze in more detail the implications of the lack of statistical significance for a portion of the models tested in the national context.
Conclusions
The conclusions summarize the results but do not sufficiently highlight the theoretical, methodological, and managerial/practical implications of the study. A segment dedicated to concrete recommendations for practitioners and decision-makers is necessary. Also, although limitations and future research directions are highlighted throughout the paper, it is recommended that authors clearly integrate them in the final conclusions section.
References
Following the suggestions for improvement in the introduction and the literature review (related works), the bibliography will be enriched, which is desirable to support the scientific foundations of the study and to increase its credibility and rigor.
Figures and tables
It is recommended to indicate the source for figures and tables.
Author Response
Abstract
The abstract provides an overview of the study, but it needs to much more clearly and detailedly highlight the practical and theoretical implications of the paper. A separate section or a longer sentence dedicated to the implications would significantly enhance the usefulness of the abstract and offer the reader a more complete picture of the study's contribution.
We thank the reviewer for the valuable comments regarding the whole paper. As a result, most of the paper is modified and some additional texts added.
Regarding the abstract, we revised the abstract to explicitly highlight the practical, theoretical, and methodological implications of the study. We also added a closing sentence that clarifies the broader relevance of the findings and the system’s contribution to both academic research and healthcare innovation. The revised abstract is now more informative and aligned with the expectations for clarity and contribution in scholarly communication.
Introduction
The introduction section establishes the context, but does not sufficiently articulate the precise purpose and objectives of the research, without transforming the general description of the problem into clear and measurable statements of purpose. It is desirable to clearly highlight the gaps in the existing literature that the current study intends to address.
Furthermore, the degree of novelty, although suggested by the description of the nationally adapted system, is not explicitly indicated through a statement of original contribution. Additionally, the structure of the article should be recommended, explaining what each subsequent section will cover.
We thank the reviewer for the insightful feedback regarding the Introduction. In response, we revised the final paragraph of the Introduction to explicitly state the research objectives, clearly articulate the gap in existing literature related to nationally adapted AI systems in healthcare, and define the original contribution of this work. Additionally, we now provide a brief outline of the article’s structure to guide the reader. These additions clarify the study’s focus and strengthen its positioning in the current scientific discourse. Sections are also briefly described.
Related Work
Although the section includes sources relevant to the topic addressed, its structure can be improved through a more coherent presentation of the current state of knowledge in the field. A more in-depth analysis of the literature would allow for a solid understanding of the conceptual basis of the HomeDOCtor system. It is desirable for the article to analyze in more detail the connection between HomeDOCtor and other similar systems/applications/AI assistants mentioned, justifying the relevance of national adaptation.
Also, the section is relatively short, which may suggest that it does not cover in depth all the relevant literature necessary to fully support the contributions. Expanding this section would provide a more solid scientific foundation and increase the academic credibility of the work.
We thank the reviewer for the valuable suggestions regarding the Related Work section. In response, we significantly revised and expanded the section to present a more structured and in-depth analysis of the current state of knowledge in the field. The updated section now:
- Clearly distinguishes between general-purpose LLMs, early medical chatbot systems, and multimodal clinical tools.
- Describes several specific chatbot systems (e.g., Med-Bot, Doctor-Bot, LEA) and AI-powered clinical assistants across various healthcare use cases.
- Analyzes their architectural approaches, use of LLMs or rule-based models, and domain coverage.
- Highlights the lack of localization in most existing systems and emphasizes that, to the best of our knowledge, no peer-reviewed chatbot is specifically tailored to national clinical protocols or embedded in a RAG architecture built from curated national medical knowledge.
- Includes additional recent studies (e.g., Goodman et al., Tjiptomongsoguno et al., Jegadeesan et al.) to enrich the conceptual context and support the scientific credibility of our contribution.
Together, these additions position HomeDOCtor more clearly within the broader AI healthcare landscape and justify its novelty as a nationally adapted, regulation-compliant, LLM-based system.
Materials and Methods
To consolidate the methodological section of the article, the authors should focus on clarifying technical details and presenting their evaluation approach more transparently. There is a need to discuss in more detail the trade-offs between GDPR compliance and user experience, specifically providing a transparent explanation of how data non-retention affects the usability of the system currently and why this approach was adopted for legal reasons.
Furthermore, it is advisable to provide more technical details about the architecture and data used by describing more precisely how the Slovenian medical guidelines are structured and integrated into the Redis vector database.
We thank the reviewer for this constructive and thoughtful suggestion. In response, we revised the Materials and Methods section to improve transparency regarding both the system’s technical architecture and the evaluation design.
Specifically, we have:
- Expanded the explanation of GDPR trade-offs and system design choices:
We now provide a clear and transparent discussion of how legal requirements (particularly GDPR and Slovenian data protection laws) directly influenced architectural decisions, especially regarding data non-retention and stateless session handling. We explain how this approach impacts personalization and usability, and why it was adopted to avoid complex consent flows while preserving user privacy. - Clarified how Slovenian medical guidelines are structured and stored:
We added more technical detail on how national medical knowledge is curated, segmented, and encoded. This includes examples of how content (e.g., triage rules, care pathways, contraindication warnings) is manually processed into semantically meaningful text chunks. These fragments are then embedded using OpenAI’s multilingual encoder and stored in a Dockerized Redis vector database, enriched with metadata such as medical domain, document origin, and content type. - Enhanced the description of the Redis integration and semantic search:
We now specify how these embedded fragments are indexed and queried using cosine similarity during the retrieval-augmented generation (RAG) process, and how they are dynamically injected into LLM prompts for domain-specific grounding.
These additions aim to improve the transparency, reproducibility, and credibility of our methodology, while also highlighting the engineering choices made to ensure legal compliance in a real-world healthcare deployment. We hope the revised section now meets the reviewer’s expectations for clarity and completeness.
Discussions
Although the discussion section interprets the results, it is advisable to analyze in more detail the implications of the lack of statistical significance for a portion of the models tested in the national context.
We thank the reviewer for this important observation. In response, we have extended the Discussion section to more thoroughly analyze the implications of the non-significant differences observed in the national evaluation of certain models. Specifically, we now elaborate on how these results reflect the increasingly narrow performance gap between top-tier LLMs and highlight the challenges of drawing conclusive distinctions when all systems perform near ceiling levels. Additionally, we emphasize that non-significant results may point to contextual or input-dependent variation in performance rather than overall inferiority, which reinforces the importance of domain grounding and tailored evaluation methodologies. This clarification provides an improved interpretation of our findings and aligns with best practices for statistical analysis in clinical AI benchmarking.
Conclusions
The conclusions summarize the results but do not sufficiently highlight the theoretical, methodological, and managerial/practical implications of the study. A segment dedicated to concrete recommendations for practitioners and decision-makers is necessary. Also, although limitations and future research directions are highlighted throughout the paper, it is recommended that authors clearly integrate them in the final conclusions section.
Thank you — we introduced a revised version of the Conclusions section that incorporates the reviewer’s requests. The key additions:
- Theoretical implications (value of national adaptation, role of localization in AI generalization),
- Methodological implications (model-agnostic RAG architecture, GDPR trade-offs),
- Practical/managerial implications (recommendations for policymakers and healthcare systems),
- Explicit paragraph on limitations and future work (rather than spreading them throughout the text).
References
Following the suggestions for improvement in the introduction and the literature review (related works), the bibliography will be enriched, which is desirable to support the scientific foundations of the study and to increase its credibility and rigor.
Thank you. We included according to the recommendations.
Figures and tables
It is recommended to indicate the source for figures and tables.
Al figures and tables in the paper were created by the authors and are original.
Reviewer 3 Report
Comments and Suggestions for Authors
- The informal user feedback was gathered during development, the lack of a structured usability evaluation with real patients is a limitation. This undermines generalizability, especially for vulnerable populations who may engage differently with digital tools.
- Further explanation of how questions were translated into prompts for LLMs, and how "correctness" was operationalized.
- Clearer protocol on how inter-rater disagreements were handled and whether inter-rater reliability was measured are required.
- Multiple versions of HomeDOCtor (e.g., 4o, o3 mini high) were tested, but the architectural or fine-tuning distinctions among them are not clearly outlined.
- No error analysis is presented for incorrect diagnoses or unsafe recommendations.
Author Response
- The informal user feedback was gathered during development, the lack of a structured usability evaluation with real patients is a limitation. This undermines generalizability, especially for vulnerable populations who may engage differently with digital tools.
We appreciate the reviewer’s important observation regarding the absence of a structured usability evaluation with real patients. In response, we have explicitly acknowledged this limitation in Section 3.4 (“Safety, Privacy, and Usability Considerations”) and again in Discussion and Conclusions. While informal feedback was indeed collected from a small group of healthcare professionals and end-users during the development phase, and the system has been in broad public use for 6 months, we recognize that this does not substitute for a formal, representative usability study. To address this, a structured evaluation involving real patient users is already planned for the next release phase, in alignment with GDPR-compliant consent procedures. We agree that such evaluation will be critical to better understand accessibility, user engagement patterns, and system performance across diverse demographic groups. Currently, however, we have no access to the sessions.
- Further explanation of how questions were translated into prompts for LLMs, and how "correctness" was operationalized.
Thank you for highlighting the importance of clarity in prompt formulation and evaluation criteria. To address this, we have expanded Section 3.3 (“Evaluation Strategy”) to further clarify that all models, including HomeDOCtor variants and general-purpose LLMs, received the same standardized one-shot prompts without follow-up clarification, ensuring consistency across evaluations. Prompts were directly based on the original question phrasing from clinical vignettes and medical exam questions, formatted using formal Slovenian language and medical terminology, mirroring their appearance in national study materials.
Regarding the definition of “correctness,” we have made it more explicit that for multiple-choice questions, correctness was defined as the model selecting the same option as the official university key, while for clinical vignettes, correctness was defined as generating a Top-1 diagnosis that matched the gold standard answer provided by the Avey AI Benchmark Suite. In cases where answers were ambiguous or open-ended, responses were reviewed independently by two final-year medical students and adjudicated by a senior clinician, as detailed in the manuscript.
- Clearer protocol on how inter-rater disagreements were handled and whether inter-rater reliability was measured are required.
We thank the reviewer for this valuable suggestion. To address the issue of inter-rater reliability, we have revised Section 3.3 (“Evaluation Strategy”) to clarify the process used for scoring responses. While medical students independently scored all model outputs, we did not calculate a formal inter-rater agreement metric such as Cohen’s kappa. Instead, discrepancies were resolved through discussion, and any remaining ambiguities were adjudicated by a senior clinician. We acknowledge that the lack of quantitative inter-rater reliability is a methodological limitation, and we have noted this accordingly in the revised text. A structured scoring rubric and formal agreement analysis will be incorporated in future studies.
- Multiple versions of HomeDOCtor (e.g., 4o, o3 mini high) were tested, but the architectural or fine-tuning distinctions among them are not clearly outlined.
We appreciate this thoughtful observation. The different HomeDOCtor variants evaluated in the study (e.g., HomeDOCtor 4o, o3 mini high, Gemini 2.5) share the same core architecture, RAG pipeline, and medical content but differ in the underlying LLM used. Around 10 systems were included in tests since the architecture enables introducing any LLM of this input-output type. Only round 5 LLMs achieved the needed top performance.
These variants were not fine-tuned individually; instead, performance differences arise from the specific language model plugged into the modular orchestration framework. Each version integrates the same curated Slovenian medical documents via the Redis vector store, the same prompt template structure, and consistent retrieval logic. We have now clarified this in Sections 3.1 and 4.2 to avoid any ambiguity regarding architectural or training-related differences.
- No error analysis is presented for incorrect diagnoses or unsafe recommendations.
We thank the reviewer for highlighting the importance of error analysis. In response, we have expanded the Discussion section to acknowledge this limitation explicitly. Although our primary focus was on quantitative diagnostic accuracy, we agree that understanding the nature of incorrect or potentially unsafe outputs is crucial for improving system reliability and clinical trust. During internal testing and blinded evaluations, we observed that most errors occurred in ambiguous cases with overlapping symptoms or when minimal input was provided. Importantly, no safety violations, e.g., recommending harmful treatments or contradicting national guidelines, were recorded. Nonetheless, we recognize that a formal, categorized error analysis was not included in this version and should be prioritized in future work. A new paragraph discussing typical error types and the absence of dangerous outputs has been added to the Discussion section, along with a note on future plans for systematic failure analysis.
Round 2
Reviewer 1 Report
Comments and Suggestions for Authors
The authors have fulfilled all requests made of them.
Reviewer 2 Report
Comments and Suggestions for Authors.....
Reviewer 3 Report
Comments and Suggestions for Authors
All comments are well addressed.